Multi-gene analysis of Symbiodinium dinoflagellates: a perspective on rarity, symbiosis, and evolution

Pochon Xavier 1 xavier.pochon@cawthron.org.nz
Putnam Hollie M. 2
Gates Ruth D. 2
1 Environmental Technologies, Cawthron Institute , Nelson , New Zealand
2 University of Hawaii, Hawaii Institute of Marine Biology , Kaneohe, HI , USA
Qian Pei-Yuan
Electronic publication date: 2014 May 20
Publication date: 2014
Volume: 2
Electronic Location ID: e394
Received 2014 Mar 4; Accepted 2014 May 3
Copyright: © 2014 Pochon et al.
Copyright year: 2014
Copyright holder: Pochon et al.
License: This is an open access article distributed under the terms of the Creative Commons Attribution License, which permits unrestricted use, distribution, reproduction and adaptation in any medium and for any purpose provided that it is properly attributed. For attribution, the original author(s), title, publication source (PeerJ) and either DOI or URL of the article must be cited.
License URL: https://creativecommons.org/licenses/by/4.0/

Keywords: Symbiosis, Chloroplast, Rarity, Evolutionary rates, Mitochondria, Nuclear, Dinoflagellate, Symbiodinium, Multi-gene analysis

Funding: NSF OCE-0752604 NSF UH EPSCoR EPS-0903833 Swiss NSF PBGEA-115118 US EPA FP917199 This research was funded by grants from NSF to RDG (OCE-0752604) and NSF UH EPSCoR (EPS-0903833), and the Swiss National Science Foundation (PBGEA-115118 to XP). HMP was supported by funds from the US EPA (FP917199). The funders had no role in study design, data collection and analysis, decision to publish, or preparation of the manuscript.

==============================
Symbiodinium, a large group of dinoflagellates, live in symbiosis with marine protists, invertebrate metazoans, and free-living in the environment. Symbiodinium are functionally variable and play critical energetic roles in symbiosis. Our knowledge of Symbiodinium has been historically constrained by the limited number of molecular markers available to study evolution in the genus. Here we compare six functional genes, representing three cellular compartments, in the nine known Symbiodinium lineages. Despite striking similarities among the single gene phylogenies from distinct organelles, none were evolutionarily identical. A fully concatenated reconstruction, however, yielded a well-resolved topology identical to the current benchmark nr28S gene. Evolutionary rates differed among cellular compartments and clades, a pattern largely driven by higher rates of evolution in the chloroplast genes of Symbiodinium clades D2 and I. The rapid rates of evolution observed amongst these relatively uncommon Symbiodinium lineages in the functionally critical chloroplast may translate into potential innovation for the symbiosis. The multi-gene analysis highlights the potential power of assessing genome-wide evolutionary patterns using recent advances in sequencing technology and emphasizes the importance of integrating ecological data with more comprehensive sampling of free-living and symbiotic Symbiodinium in assessing the evolutionary adaptation of this enigmatic dinoflagellate.

Introduction

Dinoflagellates in the genus Symbiodinium are essential components of coral reef ecosystems in their role as photosynthetic endosymbionts of a myriad of marine organisms belonging to at least five distinct phyla: Foraminifera, Porifera, Cnidaria, Mollusca, and Platyhelminthes (Trench, 1993). Although highly predominant within benthic hosts, symbiotic associations have also been reported in the pelagic medusa Cotylorhiza tuberculata (Astorga, Ruiz & Prieto, 2012). Perhaps best known for their relationship with scleractinian corals, Symbiodinium spp. underpin the productivity and calcification that creates coral skeletons and the structures known as coral reefs that serve as habitat for the immense biodiversity these coastal ecosystems support.

Research conducted during the last two decades has allowed extensive genotyping of endosymbiotic Symbiodinium in both the Western Atlantic and Indo-Pacific Oceans and across benthic host taxa at a variety of spatial and temporal scales (reviewed in Coffroth & Santos, 2005; Franklin et al., 2012; Stat, Carter & Hoegh-Guldberg, 2006). Several recent studies have also begun to describe Symbiodinium diversity in free-living environments, including the water column (Manning & Gates, 2008; Pochon et al., 2010; Takabayashi et al., 2012), sediments (Pochon et al., 2010; Porto et al., 2008; Takabayashi et al., 2012), coral sand (Hirose et al., 2008), coral rubble (Coffroth et al., 2006), on the surface of macroalgal beds (Porto et al., 2008; Venera-Ponton et al., 2010), and in fish feces (Castro-Sanguino & Sánchez, 2012; Porto et al., 2008).

Historically, the pioneering work of Rowan & Powers (1992) divided the genus Symbiodinium into three phylogenetic groups referred to as clades A-C using nuclear small subunit ribosomal (nr18S) sequences. Despite the conserved nature of this marker, sequence variation between clades is comparable to other dinoflagellate taxa placed in different orders (Rowan & Powers, 1992). Later, the use of more variable nuclear large subunit ribosomal (nr28S) sequences was applied across broader host taxa and geographic scales (Santos et al., 2002; Pawlowski et al., 2001; reviewed in Baker, 2003), and ultimately led to the molecular classification of Symbiodinium into nine lineages (Pochon & Gates, 2010), clades A through I (Table 1). Clades D, F, and G have been further divided into sub-clades D1-D2, F2-F5, and G1-G2, respectively, using nr28S and the chloroplast large subunit ribosomal DNA (cp23S) domain V (Hill et al., 2011; Pochon, LaJeunesse & Pawlowski, 2004; Pochon et al., 2006). Comparative phylogenetic reconstructions have yielded similar evolutionary relationships among Symbiodinium clades using nr28S and cp23S genes (Santos et al., 2002; Pochon & Gates, 2010), as well as when using the coding region of the plastid-encoded photosystem II protein D1 (psbA; Takishita et al., 2003), the mitochondrial cytochrome oxidase I (coI; Takabayashi, Santos & Cook, 2004), and mitochondrial cytochrome b (cob; Zhang, Bhattacharya & Lin, 2005; Sampayo, Dove & LaJeunesse, 2009). However, compared to other markers the nr28S gene typically yields better-resolved phylogenies and is therefore considered as a ‘benchmark gene’ for clade-level analysis of Symbiodinium (Pochon et al., 2012).

Table 1 Summary of existing Symbiodinium lineages.

The nine clades (A–I) and eight sub-clades (D1-D2, F2-F5, and G1-G2) that constitute the genus Symbiodinium, with selected literature highlighting the habitat prevalence/preference of each lineage.

Clade/Sub-clade	in hospite/free-living	Habitat Preferences/Prevalence	References	
A	in hospite	Cnidaria	(LaJeunesse, 2001; Reimer et al., 2006; Stat, Morris & Gates, 2008)	
	in hospite	Mollusca	(Baillie, Belda-Baillie & Maruyama, 2000; Ishikura et al., 2004; LaJeunesse et al., 2010)	
	in hospite	Plathyelminthes	(Baillie, Belda-Baillie & Maruyama, 2000)	
	free-living	Water column	(Manning & Gates, 2008; Pochon & Gates, 2010; Takabayashi et al., 2012)	
	free-living	Sediment	(Pochon & Gates, 2010; Porto et al., 2008; Takabayashi et al., 2012)	
	free-living	Reef sand/rubbles	(Coffroth et al., 2006; Hirose et al., 2008)	
	free-living	Macroalgal beds	(Porto et al., 2008)	
	free-living	Fish feces	(Castro-Sanguino & Sánchez, 2012; Porto et al., 2008)	
B	in hospite	Cnidaria	(Coffroth, Santos & Goulet, 2001; LaJeunesse, 2001; Santos, Taylor & Coffroth, 2001)	
	in hospite	Mollusca	(LaJeunesse, 2002)	
	in hospite	Porifera	(Hunter, LaJeunesse & Santos, 2007)	
	free-living	Water column	(Manning & Gates, 2008; Pochon & Gates, 2010; Takabayashi et al., 2012)	
	free-living	Sediment	(Pochon & Gates, 2010; Porto et al., 2008; Takabayashi et al., 2012)	
	free-living	Reef rubbles	(Coffroth et al., 2006)	
	free-living	Macroalgal beds	(Porto et al., 2008)	
	free-living	Fish feces	(Castro-Sanguino & Sánchez, 2012; Porto et al., 2008)	
C	in hospite	Foraminifera	(Pochon et al., 2001; Pochon et al., 2006; Pochon et al., 2007; Pochon, LaJeunesse & Pawlowski, 2004)	
	in hospite	Cnidaria	(Coffroth & Santos, 2005; LaJeunesse, 2005; Sampayo et al., 2007; Wagner et al., 2011)	
	in hospite	Mollusca	(Baillie, Belda-Baillie & Maruyama, 2000; Ishikura et al., 2004; LaJeunesse et al., 2010)	
	in hospite	Plathyelminthes	(Baillie, Belda-Baillie & Maruyama, 2000)	
	free-living	Water column	(Manning & Gates, 2008; Pochon & Gates, 2010; Takabayashi et al., 2012)	
	free-living	Sediment	(Pochon & Gates, 2010; Porto et al., 2008; Takabayashi et al., 2012)	
	free-living	Macroalgal beds	(Porto et al., 2008; Venera-Ponton et al., 2010)	
D1	in hospite	Cnidaria	(Brown et al., 2000; Correa & Baker, 2009; Jones et al., 2008)	
	in hospite	Mollusca	(Ishikura et al., 2004; LaJeunesse et al., 2010)	
	free-living	Water column	(Manning & Gates, 2008; Takabayashi et al., 2012)	
D2	in hospite	Foraminifera	(Pochon et al., 2007; Garcia-Cuetos, Pochon & Pawlowski, 2005)	
	in hospite	Porifera	(Carlos et al., 1999)	
E	in hospite	Cnidaria	(LaJeunesse & Trench, 2000; LaJeunesse, 2001)	
	free-living	Water column	(Carlos et al., 1999; Gou et al., 2003; Santos, 2004)	
F2	in hospite	Foraminifera	(Pochon et al., 2001; Pochon et al., 2006; Pochon et al., 2007; Pochon & Gates, 2010)	
	in hospite	Cnidaria	(Rodriguez-Lanetty, Cha & Song, 2002)	
F3	in hospite	Foraminifera	(Pochon et al., 2001; Pochon et al., 2006; Pochon et al., 2007; Pochon & Gates, 2010)	
F4	in hospite	Foraminifera	(Pochon et al., 2001; Pochon et al., 2006; Pochon et al., 2007; Pochon & Gates, 2010)	
F5	in hospite	Foraminifera	(Pochon et al., 2001; Pochon et al., 2006; Pochon et al., 2007; Pochon & Gates, 2010)	
G1	in hospite	Foraminifera	(Pochon et al., 2001; Pochon et al., 2006; Pochon et al., 2007; Pochon & Gates, 2010)	
G2	in hospite	Cnidaria	(Bo et al., 2011; van Oppen et al., 2005)	
	in hospite	Porifera	(Schoenberg & Loh, 2005; Schoenberg et al., 2008; Hill et al., 2011)	
	free-living	Water column	(Takabayashi et al., 2012)	
	free-living	Sediment	(Takabayashi et al., 2012)	
	free-living	Fish feces	(Castro-Sanguino & Sánchez, 2012)	
H	in hospite	Foraminifera	(Pochon et al., 2001; Pochon et al., 2006; Pochon et al., 2007; Pochon & Gates, 2010)	
	free-living	Water column	(Manning & Gates, 2008)	
I	in hospite	Foraminifera	(Pochon & Gates, 2010)	

The nine existing clades and eight sub-clades of Symbiodinium have been largely delineated based on host-symbiont associations (Table 1). For example, clades A, B, C, and D1 most commonly associate with Molluscan and Cnidarian hosts, clades B, D2, and G2 with Poriferan hosts, and clades F, G1, H, and I with Foraminifera. To date, the majority of Symbiodinium clades have also been found in the free-living environment (Table 1), particularly clades A and B which appear to contain a high number of unique strains that may be exclusively adapted to a free-living mode of life (Coffroth et al., 2006; Hirose et al., 2008; Takabayashi et al., 2012; Yamashita & Koike, 2013). However, representatives from all clades are likely to be soon characterized from the free-living environment as novel sequencing technologies now provide researchers with unprecedented screening sensitivity and ability to quickly design novel Symbiodinium-specific markers with increased resolution. To date, a number of high-resolution markers have been employed for fine-scale studies investigating the biogeography, host specificity, physiology, and ecological partitioning of specific strains within Symbiodinium clades, including microsatellite loci (Thornhill et al., 2009), the Internal Transcribed Spacer regions 1 and 2 of the nuclear ribosomal DNA (van Oppen et al., 2005; Iglesias-Prieto et al., 2004), and recently the non-coding region of psbA (LaJeunesse & Thornhill, 2011; Thornhill et al., 2014). However, none of these markers have yet been successfully employed for characterizing free-living populations of Symbiodinium due to clear challenges of specifically targeting Symbiodinium against the backdrop of the complex micro- and meio-eukaryotic diversity found in environmental samples. In an attempt to characterize novel markers for symbiotic and free-living Symbiodinium, Pochon et al. (2012) used available Expressed Sequence Tags (EST) libraries for Symbiodinium (Leggat et al., 2007; Voolstra et al., 2008), to identify 84 candidate genes, and perform in-depth phylogenetic analyses of four relatively fast evolving genes (coI, calmodulin, rad24, and actin). Other more conserved genes, including the elongation factor 2 (elf2) and the cob genes, were also recovered from EST libraries and sequenced for future clade-level analysis.

Our current understanding on the divergence pattern and evolution of Symbiodinium clades is relatively limited. A standard molecular clock using nr28S sequence data, suggested that the ancestor of the Symbiodinium species complex evolved during the K-T boundary (65 MYA) in warm tropical waters (Tchernov et al., 2004), which corresponds to a major transition time from the extinct Mesozoic rudist-based reefs, to the modern scleractinian-dominated reefs. Pochon & Pawlowski (2006) later employed a relaxed molecular clock approach with nr28S data and suggested that Symbiodinium clades started to diversify from ancestral clade A some 50 MYA, in the beginning of Eocene. Their analysis revealed that the major diversifications of clades occurred during global cooling periods: the origination of Symbiodinium clades A, B, D, E, and G during the Eocene cooling, followed by a massive radiation that took place in all lineages since mid-Miocene (15 MYA).

To improve our understanding of Symbiodinium clade evolution, in this study, we present a ‘clade-level’ multi-gene analysis incorporating samples from all known Symbiodinium clades and sub-clades (Table 2). We selected two genes from three distinct organelles (nucleus: nr28S & elf2; chloroplast: cp23S & psbA; and mitochondria: coI & cob) to test the following hypotheses: (1) single gene phylogenies will yield statistically distinct clades relationships; (2) A six-gene concatenated tree will be statistically different from benchmark nr28S; and (3) Pair-wise relative substitution rate analyses will reveal compartment-specific differences in evolutionary rates among Symbiodinium clades and organelles. Our results are integrated within the current state of knowledge of free-living and endosymbiotic Symbiodinium lineages (Table 1) and may serve as a basis for future studies investigating evolutionary implications of rarity and symbiotic/free-living lifestyles among Symbiodinium dinoflagellates.

Table 2 Description of Symbiodinium samples, host origin, and GenBank accession numbers of all DNAs used in this study.

Sample#	Cladea	ITS2b	Host origin	Isolate IDc	nr28S	elf2	cp23S	psbA	coI	cob	
1	C	C1	Amphisorus hemprichii	2359X [S]	JN558040	JN557869	JN557969	JN557844	JN557891	JN557943	
2		C90	Sorites sp.	1355X [S]	JN558045	JN557871	JN557975	JN557846	JN557893	JN557945	
3		C91	Sorites sp.	2467X [S]	JN558048	JN557872	JN557978	JN557847	JN557894	JN557946	
4		C15	Amphisorus hemprichii	2361X [S]	JN558042	JN557870	JN557972	JN557845	JN557892	JN557944	
5	H	H1	Sorites sp.	2382X [S]	JN558051	JN557873	JN557981	JN557848	JN557895	JN557947	
6		H1a	Sorites sp.	2350X [S]	JN558053	JN557874	JN557984	JN557849	JN557896	JN557948	
7	F2	F2	Sorites sp.	206J [S]	JQ247043	JQ277946	JQ247052	JQ277935	JQ277957	JQ277979	
8		F2a	Sorites sp.	215J [S]	JQ247044	JQ277947	JQ247053	JQ277936	JQ277958	JQ277980	
9	F3	F3.2	Amphisorus hemprichii	2551X [S]	JQ247046	JQ277949	JQ247055	JQ277938	JQ277960	JQ277982	
10		F3.1a	Amphisorus hemprichii	455X [S]	JQ247045	JQ277948	JQ247054	JQ277937	JQ277959	JQ277981	
11	F4	F4.1	Sorites sp.	5121X [S]	JQ247047	JQ277950	JQ247056	JQ277939	JQ277961	JQ277983	
12		F4.8	Sorites sp.	2692X [S]	JQ247048	JQ277951	JQ247057	JQ277940	JQ277962	JQ277984	
13	F5	F5.1	Meandrina meandrites	RT-133 [C]	JN558063	JN557876	JN557996	JN557851	JN557898	JN557950	
14		F5.1d	Sinularia sp.	Sin [C]	JN558069	JN557877	JN558000	JN557852	JN557899	JN557951	
15		F1	Montipora verrucosa	Mv [C]	JN558066	JN557875	JN557997	JN557850	JN557897	JN557949	
16		F5.2g	Montastraea faveolata	Mf [C]	JN558072	JN557878	JN558004	JN557853	JN557900	JN557952	
17	B	B1	Plexaura kuna	704 [C]	JN558057	JN557879	JN557991	JN557854	JN557901	JN557953	
18		B2	Eunicea flexuosa	Pflex [C]	JN558060	JN557880	JN557993	JN557855	JN557902	JN557954	
19		B19a	Plexaura kuna	703 [C]	JN558055	JN557881	JN557987	JN557856	JN557903	JN557955	
20	I	I1	Sorites sp.	OHU7 [S]	FN561559	JQ277955	FN561563	JQ277944	JQ277966	JQ277988	
21		I2	Sorites sp.	OHU3 [S]	FN561560	JQ277956	FN561564	JQ277945	JQ277967	JQ277989	
22	D1	D1	Acropora sp.	A001 [C]	JN558075	JN557882	JN558007	JN557857	JN557904	JN557956	
23		D1a	unknown anenome	Ap02 [C]	JN558078	JN557883	JN558010	JN557858	JN557905	JN557957	
24	D2	D1.1	Marginopora vertebralis	2485X [S]	JQ247049	JQ277952	JQ247058	JQ277941	JQ277963	JQ277985	
25		D1.2	Haliclona koremella	HK [C]	JN558081	JN557884	JN558013	JN557859	JN557906	JN557958	
26	G1	G2	Marginopora vertebralis	2479X [S]	JN558089	JN557885	JN558019	JN557860	JN557907	JN557959	
27		G2b	Marginopora vertebralis	3590X [S]	JN558088	N/A	JN558017	JN557861	JN557908	JN557960	
28	G2	G2.1*	Cliona orientalis	OR2 [S]	JQ247050	JQ277953	JQ247059	JQ277942	JQ277964	JQ277986	
29		G2.2*	Cliona orientalis	RN3 [S]	JQ247051	JQ277954	JQ247060	JQ277943	JQ277965	JQ277987	
30	E	E1	Anthopleura elegantissima	RT-383 [C]	JN558084	N/A	JN558015	JN557862	JN557909	JN557961	
31	A	A2_ 1	Bartholomea annulata	RT-23 [C]	JN558097	JN557887	JN558029	JN557864	JN557911	JN557963	
32		A2_ 2	Gorgonia ventallina	RT-89 [C]	JN558100	JN557888	JN558032	JN557865	JN557912	JN557964	
33		A3	Pseudoplexaura porosa	725 [C]	JN558091	JN557889	JN558021	JN557866	JN557913	JN557965	
34		A13	Plexaura kuna	708 [C]	JN558094	JN557886	JN558027	JN557863	JN557910	JN557962	
Outgroup1	G. simplex	N/A	N/A	CCMP419 [C]	JN558103	JN557890	JN558033	JN557867	JN557914	JN557966	
Outgroup2	P. beii	N/A	N/A	PB-1 [C]	JN558106	N/A	N/A	N/A	JN557915	JN557967	
Outgroup3	P. glacialis	N/A	N/A	CCMP1383 [C]	JN558108	N/A	JN558036	JN557868	JN557916	JN557968	
Notes.

a Letters A to H refer to the Symbiodinium clades, and lineages D1-D2, F2-F5, and G1-G2 are the Symbiodinium sub-clades.

b Alpha-numeric names correspond to Symbiodinium ITS2 rDNA molecular taxonomy sensu Pochon et al. (2007). Letters correspond to the Symbiodinium clades, and numbers correspond to a specific ITS2 sequence. All samples are genetically distinct, except for Symbiodinium A2, which was found in two distinct cultures and referred here to as A2_ 1 and A2_ 2. Types D1.1 and D1.2 corresponds to the symbionts of the foraminifer M. vertebralis and the sponge Haliclona koremella, respectively (see Pochon et al., 2007 for details), and were previously described as belonging to Symbiodinium sub-clade D1 (Garcia-Cuetos, Pochon & Pawlowski, 2005; Pochon & Pawlowski, 2006), but reclassified here as sub-clade D2. Sub-clade D1 contains Symbiodinium strains that are commonly associated with Scleractinian corals, such as symbiont ITS2 types D1 and D1a (Stat & Gates, 2011). Types G2 and G2b belong to sub-clade G1 as shown in Pochon et al. (2012).

c Samples ID are followed by [C] if DNA was extracted from a culture, or [S] if extracted from a symbiotic host. All GenBank accession numbers starting with the letters ‘JQ’ were obtained in the present study.

* Indicates new ITS2 sequences; novel types G2.1 and G2.2 belong to sub-clade G2 following Hill et al. (2011).

Materials and Methods

DNA samples

Thirty-four DNA samples encompassing all known Symbiodinium clades (A-I) and sub-clades (F2-F5; D1-D2; G1-G2) were selected for phylogenetic analyses (Table 2). These samples included fifteen axenic Symbiodinium cultures belonging to five clades/sub-clades (A, B, D, E, and F5), seventeen samples originally isolated from symbiotic soritid foraminifera (Pochon et al., 2007; Pochon & Gates, 2010) belonging to six Symbiodinium clades/sub-clades (C, D2, F2-F4, G1, H, and I), and two samples extracted from the symbiotic bioeroding sponge genus Cliona and belonging to Symbiodinium sub-clade G2 (see Bo et al., 2011; Hill et al., 2011). Additionally, three cultured dinoflagellates, Gymnodinium simplex [CCMP 419], Pelagodinium beii (Siano et al., 2010), and Polarella glacialis [CCMP 1383] were used as outgroups in our analyses following Pochon et al. (2012).

Genes selection, DNA extraction and sequencing

Six genes from three organelles were chosen for phylogenetic analyses. These include two nuclear genes (1) nr28S (D1-D3 region) [920 base-pairs] and (2) elf2 [473 bp]; two chloroplast genes (3) cp23S (Domain V) [647 bp] and (4) the coding region of psbA [700 bp]; and two mitochondrial genes (5) coI [1057 bp] and (6) cob [906 bp]. Sequences for analysis were gathered from 26 samples obtained from a previous study (Pochon et al., 2012), nine DNA samples were extracted and partially analyzed in other studies (Pochon et al., 2007; Pochon & Gates, 2010) and further sequenced here to cover all genes using the primers and PCR cycling conditions described in Pochon et al. (2012), and two DNA samples were extracted from sponge tissues of the genus Cliona (courtesy of C. Schoenberg) and sequenced for all genes following Pochon et al. (2012) (see Table 2). The psbA gene was not reported in Pochon et al. (2012) and was PCR amplified in this study using the forward primer psbA_1.0 (5′-CWGTAGATATTGATGGWATAAGAGA-3′) located at the 5′ end of the coding region and the reverse primer psbA_3.0 (5′-TTGAAAGCCATTGTYCTTACTCC-3′) located approximately 700 bp downstream from the 5′ end and using standard thermocycling conditions with an annealing temperature of 52 °C. All sequences were obtained by direct sequencing, except for nr28S and cp23S sequences, which were cloned prior to sequencing in Pochon et al. (2012), and a single sequence per sample included in the present study. In all cases, the variability between cloned sequences of any given sample was minimal (e.g., see Figure S1 of Pochon et al., 2012), ranging between 0 and 4 bp difference (data not shown). However, sequences showing the shortest branch length in each sample were selected (data not shown). In cases where several sequences showed the same short branch length, one sequence was randomly chosen among them and included in the analysis.

Phylogenetic analyses

DNA sequences were inspected and assembled using Sequencher v4.7 (Gene Codes Corporation, Ann Arbor, MI, USA) and manually aligned with BioEdit v5.0.9 sequence alignment software (Hall, 1999). Thirteen distinct DNA alignments were generated: six alignments corresponding to individual gene alignments, one fully concatenated alignment of all six genes (ALL Concat), and six partially concatenated alignments including all possibilities of five genes each (i.e., each alignment excluded one of the six gene candidates). Concatenated alignments were created using the ‘join sequence files’ option in TREEFINDER v12.2.0 (Jobb, von Haeseler & Strimmer, 2004). elf2 was included in these analyses despite two missing samples (see samples #27 and #30; Table 2), which were coded as missing data in all concatenated alignments. GenBank accession numbers for all investigated sequences are shown in Table 2.

Each DNA alignment was analyzed independently under both Maximum-likelihood (ML) and Bayesian environments. Best-fit models of evolution were estimated for each alignment (see Table S1) using Modeltest v3.7 (Posada & Crandall, 1998). ML analyses were carried out using PhyML v3.0 (Guindon et al., 2009), and the reliability of internal branches was assessed using 100 bootstraps with subtree pruning-regrafting branch swapping. Bayesian tree reconstructions with posterior probabilities were inferred using MrBayes v3.2 (Ronquist et al., 2012), using the same model of DNA evolution as for the ML analyses. Four simultaneous Markov chains were run for 1,000,000 generations with trees sampled every 10 generations, with 50,000 initial trees discarded as “burn-in”, based on visual inspections. Concatenated alignments were run under ML and Bayesian environments as described above, with the alignments partitioned so that the specific model of evolution corresponded to each gene fragment.

Topological tests, rate calculations, and statistical analyses

To compare the topology of the various trees, approximately unbiased (AU) topological congruency tests (Shimodaira, 2002) were performed using site likelihood calculation in RaxML v7.2.5 (Stamatakis, 2006), followed by AU tests using CONSEL (Shimodaira & Hasegawa, 2001) with default scaling and replicate values. elf2 was excluded from the single gene analyses due to missing data (samples #27 and #30; Table 2), but was included in the concatenated analyses (see above).

In order to determine evolutionary rates among Symbiodinium lineages for each of the six investigated genes, relative-rate tests (RRT) were performed using the program RRTREE v1.1 (Robinson-Rechavi & Huchon, 2000). Clades and sub-clades were compared in a pair-wise fashion with G. simplex as the outgroup. Relative rates of evolution (K-scores from RRTREE analysis above) were compared among clades and among cellular organelles using a two way ANOVA, followed by post hoc analysis with Tukey’s Honestly Significant Difference (THSD) test.

Results

DNA alignments for the six investigated genes ranged between 473 (elf2) and 1,057 bp (coI). Individual phylogenies were generated (Fig. 1), and each was compared to the topology obtained with the nr28S gene, which is the current molecular taxonomic benchmark for the clade-level classification of Symbiodinium (Hill et al., 2011; Pochon & Gates, 2010; Pochon et al., 2012). Overall, the cladal relationships were remarkably similar among the genes investigated, particularly the basal positions of clades A, D, E and G, and the derived positions of clades B, C, F, H, and I. Symbiodinium clades were relatively well resolved in the nuclear and chloroplastic genes, but not the mitochondrial genes, which placed clades C, F, and H in completely unresolved monophyletic groups (see Figs. 1E and 1F). However, with the exception of nr28S, the relationships amongst clades were weakly supported for all markers, especially in the higher parts of the trees, and this was particularly evident for psbA where relationships between clades B, C, D, F, G, H, and I were completely unresolved (Fig. 1D). Furthermore, the relationships between sub-clades within clades D, F, and G showed contrasting results. Well-supported monophyly of all sub-clades was only observed in the nr28S gene (Fig.1A). Notably however, clade G sub-clades (G1 and G2) formed a monophyletic group across all genes. In contrast, the monophyly of clade F and clade D sub-clades was only resolved with nr28S (Fig. 1A) and nr28S and cob (Figs. 1A and 1F), respectively. All Symbiodinium strains belonging to the same sub-clade grouped together across all genes, with two noteworthy exceptions. First, the four samples of sub-clade F5 (#14-16) separated into two groups in cob (Fig. 1F). Second, sample #24 (Table 2) of sub-clade D2 diverged significantly to the root of the tree in cp23S (Fig. 1C).

Figure 1 Single-gene phylogenies of Symbiodinium using two genes from three organelles.

Best Maximum likelihood (ML) topologies for Symbiodinium clades and sub-clades A to I based on the nuclear genes (A) nr28S and (B) elf2, the chloroplastic genes (C) cp23S and (D) psbA, and the mitochondrial genes (E) coI and (F) cob. Numbers in brackets refer to the Symbiodinium strains detailed in Table 2. Numbers at nodes represent the ML bootstrap pseudoreplicate (BP) values (underlined numbers; 100 BP performed) and Bayesian posterior probabilities (BiPP). Black dots represent nodes with <95% BP and BiPP of 1.0. Nodes without numbers correspond to BP and BiPP lower than 70% and 0.8, respectively. Nodes displaying BP lower than 50% were manually collapsed. The phylograms were rooted using the dinoflagellates Gymnodinium simplex, Pelagodinium beii, and/or Polarella glacialis. GenBank accession numbers are given in Table 2. Note: All clades are represented, except for clade E in the elf2 phylogeny.

In order to increase the phylogenetic signal and assess which of the individual markers best reflects the most well resolved evolutionary history of Symbiodinium, a series of gene concatenation analyses were conducted. In total, seven distinct concatenated alignments were analyzed, including one fully concatenated alignment of all six genes (ALL Concat) consisting of a total length of 4,703 bp, and six partially concatenated alignments ranging in length from 3,646 bp (ALL except coI) and 4,230 bp (ALL except elf2), and including all possibilities of five genes each (see Methods). Phylogenetic analysis of the fully concatenated dataset (ALL Concat, Fig. 2) resulted in a highly resolved Symbiodinium tree with identical topology to nr28S gene, but with much stronger phylogenetic signal as evidenced by a significant increase in statistical support at all nodes (Fig. 2). Other concatenated alignments yielded weaker nodes support and unstable cladal relationships globally (data not shown).

Figure 2 Best topology of Symbiodinium based on six concatenated genes.

Maximum likelihood (ML) topology for Symbiodinium clades and sub-clades A to I based on fully concatenated DNA alignment (ALL Concat; 4,703 bp) of all six genes investigated in this study. The Symbiodinium strains within each clade/sub-clade are referred using the specific numbers and corresponding ITS2 names in brackets (Table 2, Fig. 1). Numbers at nodes represent the ML bootstrap pseudoreplicate (BP) values (underlined numbers; 100 BP performed) and Bayesian posterior probabilities (BiPP). Black dots represent nodes with 100% BP and BiPP of 1.0. The phylograms were rooted using the dinoflagellates Gymnodinium simplex, Pelagodinium beii, and Polarella glacialis.

Approximately unbiased (AU) topological congruency tests (Shimodaira, 2002) were used to verify whether any of the distinct phylogenies resulted in statistically identical topologies. First, pair-wise comparisons of single gene phylogenies (Fig. 1) resulted in significant p-values (p < 0.05) in all cases, indicating that the different genes have not followed identical evolutionary trajectories (see Table S2A). Second, concatenated topologies tested against single gene topologies, also resulted in significant p-values in all instances (data not shown). Third, pair-wise comparisons of single gene phylogenies to the concatenated topologies, revealed that the two longest genes, coI and nr28S, resulted in 5 and 6 significant topological comparisons, respectively (see Table S2B). Despite the relatively smaller size of nr28S (920 bp) compared to coI (1057 bp), nr28S was the only marker yielding a statistically identical topology to the fully concatenated topology (ALL Concat). The nr28S topology, however, was not identical to the best topology of the concatenated alignment excluding the nr28S gene fragment (see ALL except nr28S in Table S2B). Similarly, pair-wise comparisons of concatenated topologies revealed that significant p-values (p < 0.05) were only observed against the ‘ALL except nr28S’ topology (Table S2B).

Figure 3 Comparison of relative rates of evolution among Symbiodinium organelles and clades.

Plot of mean relative rates of evolution (mean ± sem) across the (A) three organelles and (B) nine clades. Lower case, italicized letters above the bars represent post hoc THSD tests with significant differences between (A) the three organelles and (B) between clades (groups of three bars). Sample sizes are shown at the base of each bar, except clade F, where for each bar n = 20.

The variable branch lengths observed in the six phylograms (Fig. 1) are directly proportional to the amount of character change; hence the longest branches are indicative of increased evolutionary rates of any given Symbiodinium strain. In most cases, increased rates of Symbiodinium clades/sub-clades appeared to be gene-specific rather than a character state maintained across all markers. K-scores from relative rate tests were coupled with ANOVA to compare the relative rates of evolution among the clades and organelles (Fig. 3) examining all clades across the three makers. There was no significant interaction of clade and organelle (F16,175 = 1.57, p = 0.081), indicating that the pattern of changes in rates of evolution among clades were similar across organelles. Overall the general pattern of slower relative rates of evolution for some of the basal clades (A, E) and faster rates in more derived clades (C, F, H, and I) is held across organelles. However, organelles differed in their relative rates of evolution (F2,175 = 248.9, p = 0.0001), driven by rapid rates in the chloroplastic and nuclear compartments in comparison to the mitochondrial compartment (Fig. 3A), with the most rapid rates found in the chloroplastic markers due the high evolutionary rates of clade I and sub-clade D2 (see Figs. 1C and 1D). Additionally, there was a significant difference between clades (F8,175 = 3.87, p = 0.0003) driven by the slow rates of clade A, and the rapid rates of clade I (Fig. 3B).

Discussion

Multi-gene analysis supports nr28S as a benchmark lineage marker

Our knowledge of Symbiodinium evolution has historically been constrained by the limited number of phylogenetic markers that have been applied to this group. To date, less than 15 DNA loci have been used to examine Symbiodinium diversity in a phylogenetic context (LaJeunesse & Thornhill, 2011; Pochon et al., 2012; Rowan & Powers, 1992; Sampayo, Dove & LaJeunesse, 2009; Takabayashi, Santos & Cook, 2004; Takishita et al., 2003; van Oppen et al., 2001), and evolutionary relationships among all existing Symbiodinium lineages have never been inferred using more than two concatenated genes (Pochon & Gates, 2010). This study is the first to perform a multi-gene analysis using six markers representing three cellular organelles and integrating biological samples from all known clades and selected sub-clades that encompass the genus Symbiodinium. In spite of the overall similarity among the trees for each nuclear, chloroplastic and mitochondrial gene (Fig. 1), their topologies were statistically different (Table S2). This reflects within and among clade differences inherent to the individual markers. Most notably being the unstable positions of clades D, E, F5 and H, as well as weak support for among clade relationships observed in most markers investigated. Long-branch attraction artifacts (Felsenstein, 1985) most likely accounted for the placement of sub-clade D2 (sample #24) at the root of the tree in the chloroplast 23S topology, and for the monophyly of samples #7, 8, 13, and 14 in the cob topology. While the markers investigated here are conserved genes that have a priori limited utility for finer scale (i.e., within clade) analysis, each contains a unique set of characteristics, including variable cladal resolution and/or evolutionary rates (e.g., see samples #2 and #3 in coI or samples #7, 8, 13, 14 in cob), hence each marker has the potential to address different questions. These differences thus support our previous conclusion that no one gene fits all of the taxonomic questions being asked in the genus Symbiodinium (Pochon et al., 2012).

Our fully concatenated analysis, incorporating all investigated genes and totaling 4,703 bp, resulted in a highly resolved phylogeny that was statistically identical to the nr28S gene, a gene used as the benchmark for assigning Symbiodinium lineages (Fig. 2; Table S2). The fact that the concatenated nuclear, chloroplastic, and mitochondrial genes display overall similar evolutionary histories, suggests that the molecular taxonomy of the currently recognized Symbiodinium clades using nr28S is robust (Pochon et al., 2006; Pochon & Gates, 2010), and that the points of clade differentiation are ancient, allowing for a concerted evolution of these conserved genes across genomes. These new results support a sequential evolution of Symbiodinium clades A/E/G1-G2/D1-D2/I/B/F2-F5/H/C, from most ancestral to most derived, respectively. It appears that there is a level of constraint in the system, with recombination likely being a rare event (Santos & Coffroth, 2003; but see Chi, Parrow & Dunthorn, 2014), a feature that maintains separation among lineages.

Compartment specific evolution and link to environmental preference/prevalence

Dinoflagellates are characterized by several genetic distinguishing features, including large genome size, and complex architecture and gene regulation (Barbrook et al., 2010; Hackett et al., 2004; Howe, Nisbet & Barbrook, 2008). One prominent feature is the large number of genes that have relocated from the ancestral organellar genome to the nucleus, resulting in a significant reduction in plastid and mitochondrondrial genomes. For example, the few genes that remain in the plastid of peridinin-containing dinoflagellates are primarily the core subunits of the photosystem (including cp23S), and the cytochrome b6f and ATP synthase complex (about 16 genes including psbA) (Hackett et al., 2004). Similarly, the mitochondrial genome of dinoflagellates has been reduced to three protein-coding genes (coI, coIII, and cob), but also contains a large number of non-functional fragments separated by repetitive non-coding DNA (Barbrook et al., 2010; Waller & Jackson, 2009). Despite the fact that the six Symbiodinium genes investigated here are only a very small subset of the Symbiodinium genome, they are physically separated in three cellular compartments, each with distinct evolutionary constraints and potential. For example, our comparisons of evolutionary rates between markers revealed that the differences among cellular compartments was primarily driven by the dissimilarity in the rates of evolution in cp23S and psbA in Symbiodinium lineages D2 and I (Figs. 1 and 2).

A possible explanation is that the increased evolutionary rates reflect rarity and adaptation to marginal habitats. It has been posited that rare taxa are important in driving evolutionary trajectories and innovations (Holt, 1997). Rarity in terms of small population size and isolation can drive high rates of adaptation and speciation (e.g., peripheral speciation; Mayr, 1963), as mutations in rare species are more likely to accumulate in the periphery of the founding population’s habitat where rare species may be subjected to persistent directional selection in the absence of gene flow, as they colonize new areas (Garcia-Ramos & Kirkpatrick, 1997). Such a scenario is supported by the fact that lineages D2 and I have only been documented on few occasions (Carlos et al., 1999; Pochon et al., 2007; Pochon & Gates, 2010), despite numerous Symbiodinium surveys conducted over the last 20 years in both the Western Atlantic and Indo-Pacific Oceans targeting a diversity of host taxa, as well as free-living communities, and crossing a variety of spatial and temporal scales (reviewed in Coffroth & Santos, 2005; Stat, Carter & Hoegh-Guldberg, 2006). In addition, Symbiodinium D2 and I have only been detected in the Hawaiian Archipelago and Micronesia (Guam and Palau), some of the most isolated island groups in the world and areas known for harboring high levels of endemism in marine biodiversity (Hughes, Bellwood & Connolly, 2002; Pauley, 2003). Both lineages have been suspected to either be free-living because of the manner in which the sample was isolated (Carlos et al., 1999), or recently ingested free-living strains due to their apparent rarity in nature (Pochon & Gates, 2010).

The high rates of evolution in chloroplastic genes in Symbiodinium sub-clade D2 and clade I might also reflect a relatively recent transition from free-living to symbiotic lifestyles. These habitats are extremely different in nature and composition, with free-living environments exhibiting high levels of environmental variability and unpredictability, while symbiotic habitats are relatively more predictable being spatially constrained and influenced by the biology of the host. These environmental differences undoubtedly drive the very different morphologies of Symbiodinium found in these two habitats, with free-living Symbiodinium flagellated and motile, and symbiotic Symbiodinium encysted and immotile. In terms of evolutionary trajectories, such differences in environment must exert a profound influence. Symbiodinium strains evolving predominantly in symbiosis must have adapted particular biochemical and chloroplastic functions in an environment that bears little or no resemblance to a free-living setting. Previous studies on the transition between symbiotic and free-living habitat show that changes in evolutionary rate occur in bacteria that have transitioned from free-living to a symbiotic lifestyle and mutualism (Lutzoni & Pagel, 1997; Moran, 1996). In addition, in some ectomycorrhizal assemblages, changes in evolutionary rate correspond to reversing from symbiotic to free-living lifestyle (Hibbett, Gilbert & Donoghue, 2000). Further, rapid and extreme environmental changes may favor the survival of rare and transitioning species, as their existing phenotypic diversity may contain traits pre-adapted to a changing environment (Holt, 1997).

Our examination of evolutionary rates for multiple markers and organelles provides an opportunity to begin addressing the implications for gene and genome evolution due to symbiotic lifestyle and dissimilarities in organellar genome constraints. Here we see the significantly slower evolutionary rates of Symbiodinium clade A compared to other clades as well as overall slower relative rates of the mitochondrial compartment across all clades (Fig. 3). As recently highlighted by Decelle (2013), the predominance of a particular lifestyle in marine microalgae (i.e., symbiotic versus free-living) can have important implications in genome evolution. Symbiodinium clade A is a basal lineage known to date back to at least 50 MYA and which has possibly survived through the climatic vicissitudes of the K-T boundary (Tchernov et al., 2004; Pochon et al., 2006). This clade easily overgrows other Symbiodinium clades in culture (Carlos et al., 1999) and shows attributes of parasitism in scleractinian corals (Stat, Morris & Gates, 2008). Additionally, clade A contains a high number of unique strains that may never establish symbiotic relationships (e.g., Coffroth et al., 2006; Hirose et al., 2008; Yamashita & Koike, 2013), and evolves at a similar rate to its close pelagic dinoflagellate relatives, contrasting with all other Symbiodinium clades which on average evolve six times faster based on nr18S sequence analyses (Shaked & de Vargas, 2006). As discussed by Decelle (2013) these differential traits and pressures of clade A, such as prevalence in the free-living environment with an occasional symbiotic lifestage (i.e., planktonic symbionts) provide a situation where the genomes are primarily influenced by external environmental pressures rather than host controlled traits. The resulting pressures are more likely to establish sexual exchanges within larger free-living populations, minimizing genomic impacts with often comparatively slower rates of evolution. In contrast, lineages that spend most of their lifecycle in hospite, which is arguably the case for most Symbiodinium clades (Table 1), tend to develop a certain dependence on the host which can lead to comparatively higher rates of change due to genome reduction and higher genetic drift associated with the absence of purifying selection through sexual recombination (Lynch, Koskella & Schaack, 2006).

Our analysis of relative-rates of evolution also indicated that mitochondrial genes evolved approximately twice slower than nuclear and chloroplastic genes. This result appears to contrast markedly with the recent study of Roy-Smith & Keeling (2012), which showed that silent site divergence of the mitochondrial genome in other protists with secondary red-algal derived plastids evolve 5–30 times faster than the divergence of their plastid genomes. These contrasting results may in part be due to the differences in DNA bases of a few selected genes in our study in comparison to the silent site divergence of complete mitochondrial and plastid genomes in Roy-Smith & Keeling (2012). Nevertheless, as there is evidence that our results from a subset of genes matches those of land plants and green algae, with more rapid rates of divergence in the plastid organelle, additional work is needed to further explore the implications of transitions between the free-living and symbiotic state for Symbiodinium, with a goal of gaining a more comprehensive understanding of the dynamics and mechanisms behind the different evolutionary trajectories observed in this study. Additionally, the increasing use of next-generation sequencing for characterizing entire Symbiodinium genomes (e.g., Barbrook, Voolstra & Howe, 2014) is an exciting avenue that provides unprecedented opportunities for the investigation of novel markers and paves the way for much more comprehensive phylogenomics studies to come.

Conclusions

Our study examines the performance of six genetic markers from three organelles in samples representing all currently documented lineages of Symbiodinium. As such it represents a comprehensive phylogenetic reconstruction of Symbiodinium, and highlights differences in the taxonomic resolution of each marker and their relative value in addressing a variety of evolutionary questions. Our series of phylogenetic analyses were conducted to address three working hypotheses. Despite striking similarities among the single gene phylogenies from distinct cellular compartments, none were evolutionarily identical confirming our first hypothesis. This result reflected within and among clade differences inherent to the individual markers. Our second hypothesis, however, was rejected and showed that a supermatrix tree incorporating all investigated genes (4,703 bp alignment) resulted in a highly resolved phylogeny that was statistically identical to the nr28S gene. This result provides additional support for the use of nr28S as a ‘clade-level’ benchmark gene for Symbiodinium. Finally, compartment-specific differences in evolutionary rates among Symbiodinium clade and gene organelle were revealed confirming our third hypothesis. Highest evolutionary rates were observed within the chloroplastic compartment, a pattern that was largely driven by fast evolving Symbiodinium clades D2 and I, two lineages that are rare in nature and which may be transitioning between free-living and symbiotic states. As such, rarity appears to associate with evolutionary innovation in a key functional compartment in Symbiodinium. The identification of different evolutionary trajectories in chloroplast genes that link with habitat and prevalence suggests that this organellar compartment is evolutionarily plastic and responsive. This finding may have important implications for our understanding of evolutionary processes that underpin a symbiotic lifestyle in this essential dinoflagellate group. Our analysis further revealed that investigated mitochondrial genes evolved approximately twice slower than nuclear and chloroplastic genes, an observation that contrasts with comparatively fast mitochondrial rates previously documented in non-symbiotic protists with secondary red-algal derived plastids. Together these results further highlight the need for deeper genome sequencing for a variety of Symbiodinium taxa with rapidly advancing next-generation sequencing approaches to understand the evolution of these enigmatic yet critical symbionts.

Supplemental information

Figure S1 Selected model of evolution and corresponding parameters for each DNA alignment used in this study

Click here for additional data file.

Table S2 Summary of Approximately Unbiased (AU) topological congruency tests performed between each DNA alignment and best ML topology

For each comparison, Table A and B shows the log likelihood difference and AU test p-value in brackets. *Accepted topologies display a p-value >0.05 (highlighted in grey). (A) Comparisons of single gene DNA alignments to single gene topologies. Elongation Factor 2 (elf2) is missing from these calculations due to missing data (missing sample #27 and #30). (B) Comparisons of single gene and concatenated DNA alignments to the concatenated topologies. elf2 was included in the concatenated alignments, where sample #27 and #30 were coded as missing data.

Click here for additional data file.

We are grateful to Fabien Burki for his guidance on the AU topological tests. We also thank Christine Schoenberg for providing us with Cliona orientalis samples, Mary-Alice Coffroth for the Symbiodinium cultures, and Colomban de Vargas and Ian Probert for the Pelagodinium beii culture, and three anonymous reviewers for their constructive comments. Most phylogenetic analyses were performed via the Bioportal computer service (http://www.bioportal.uio.no) at the University of Oslo, Norway. This is SOEST contribution number 9119 and HIMB contribution number 1588.

Additional Information and Declarations

Competing Interests

Author Contributions

DNA Deposition

The authors declare there are no competing interests. Xavier Pochon is an employee of The Cawthron Institute.

Xavier Pochon conceived and designed the experiments, performed the experiments, analyzed the data, wrote the paper, prepared figures and/or tables, reviewed drafts of the paper.

Hollie M. Putnam conceived and designed the experiments, analyzed the data, wrote the paper, prepared figures and/or tables, reviewed drafts of the paper.

Ruth D. Gates conceived and designed the experiments, contributed reagents/materials/analysis tools, reviewed drafts of the paper.

The following information was supplied regarding the deposition of DNA sequences:

All sequences have been deposited in GenBank. A detailed table is provided in the manuscript. All sequence accession numbers starting with “JQ” are novel:

JN558040 JN557869 JN557969 JN557844 JN557891 JN557943

JN558045 JN557871 JN557975 JN557846 JN557893 JN557945

JN558048 JN557872 JN557978 JN557847 JN557894 JN557946

JN558042 JN557870 JN557972 JN557845 JN557892 JN557944

JN558051 JN557873 JN557981 JN557848 JN557895 JN557947

JN558053 JN557874 JN557984 JN557849 JN557896 JN557948

JQ247043 JQ277946 JQ247052 JQ277935 JQ277957 JQ277979

JQ247044 JQ277947 JQ247053 JQ277936 JQ277958 JQ277980

JQ247046 JQ277949 JQ247055 JQ277938 JQ277960 JQ277982

JQ247045 JQ277948 JQ247054 JQ277937 JQ277959 JQ277981

JQ247047 JQ277950 JQ247056 JQ277939 JQ277961 JQ277983

JQ247048 JQ277951 JQ247057 JQ277940 JQ277962 JQ277984

JN558063 JN557876 JN557996 JN557851 JN557898 JN557950

JN558069 JN557877 JN558000 JN557852 JN557899 JN557951

JN558066 JN557875 JN557997 JN557850 JN557897 JN557949

JN558072 JN557878 JN558004 JN557853 JN557900 JN557952

JN558057 JN557879 JN557991 JN557854 JN557901 JN557953

JN558060 JN557880 JN557993 JN557855 JN557902 JN557954

JN558055 JN557881 JN557987 JN557856 JN557903 JN557955

FN561559 JQ277955 FN561563 JQ277944 JQ277966 JQ277988

FN561560 JQ277956 FN561564 JQ277945 JQ277967 JQ277989

JN558075 JN557882 JN558007 JN557857 JN557904 JN557956

JN558078 JN557883 JN558010 JN557858 JN557905 JN557957

JQ247049 JQ277952 JQ247058 JQ277941 JQ277963 JQ277985

JN558081 JN557884 JN558013 JN557859 JN557906 JN557958

JN558089 JN557885 JN558019 JN557860 JN557907 JN557959

JN558088 JN558017 JN557861 JN557908 JN557960

JQ247050 JQ277953 JQ247059 JQ277942 JQ277964 JQ277986

JQ247051 JQ277954 JQ247060 JQ277943 JQ277965 JQ277987

JN558084 JN558015 JN557862 JN557909 JN557961

JN558097 JN557887 JN558029 JN557864 JN557911 JN557963

JN558100 JN557888 JN558032 JN557865 JN557912 JN557964

JN558091 JN557889 JN558021 JN557866 JN557913 JN557965

JN558094 JN557886 JN558027 JN557863 JN557910 JN557962

JN558103 JN557890 JN558033 JN557867 JN557914 JN557966

JN558106 JN557915 JN557967

JN558108 JN558036 JN557868 JN557916 JN557968

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
