# Peer review of "Multi-gene analysis of Symbiodinium dinoflagellates: a perspective on rarity, symbiosis, and evolution"

_PeerJ, doi:10.7717/peerj.394_

## Round 0.1 · original submission · Major Revisions

· Academic Editor

Major Revisions

Dear Authors,
Two reviewers raised a number of issues that have to be carefully and adequately addressed before this paper can be considered further. Please go through the comments carefully and if you think you can address the concerns and issues (or rebuttal to some of critical points), you are welcome to send your ms back for further consideration (with additional review).

Reviewer 1 ·

Basic reporting

The dinoflagellate Symbiodinium is one of the most extensively studied phytoplankton taxa because of its ecological and economical importance. Symbiodinium drives whole reef ecosystems worldwide, and is therefore a major focus to assess impacts of climate change and man-made perturbations. Thus, it is of high importance to study the evolution of this enigmatic single-celled organism, and more particularly to have a reference phylogeny. The study of Pochon et al is very interesting in different aspects. First, they generated for the first time a multi-gene concatenated phylogeny for Symbiodinium with all the existing clades, which, paradoxically, has not been achieved before. They retrieved the same but a more robust topology as with the 28S rRNA. They therefore provide the best and updated evolutionary picture of Symbiodinium, a reference that will be used and cited in many studies.
They also carefully compare single-gene phylogenies from 3 different cellular compartments, an analysis that is rarely performed for other taxa, unfortunately. Finally, the comparison between evolutionary rates across all Symbiodinium clades is very interesting and brings new perspectives. Thus, I believe this study deserves publication in PeerJ.

Experimental design

A significant effort was done for retrieving the 6 genes for all clades of Symbiodinum, that are either from cultures, or more challenging from a symbiotic stage (in hospite) in foraminifera and sponges.

Validity of the findings

Phylogenetic analyses and statistical tests are adequate and robust. GenBank accession numbers of sequences are available and clearly presented.

Additional comments

I would have some comments and suggestions for the authors.

1) In the introduction, it would be nice to mention that Symbiodinium is also found in symbiosis with pelagic medusa (like Cotylorhiza tuberculata), and not only in benthic organisms, although highly predominant in the later case.

2) To my knowledge, genotyping symbiotic Symbiodinium has been done only/mainly on benthic hosts and not on pelagic hosts.
line 36: add "benthic" : "across benthic host taxa".

3) In table 1 (which is very useful), it would nice to add another column before the "Habitat preference.." one, indicating whether the sequence has been found in hospite or free-living. I know that the information is already provided with "water column" or "sediment" but it would be easier to see rapidly the free-living category across different clades. The reader will see whether free living Symbiodinium sequences have been found more in a certain clade or are evenly distributed in the nine clades.

4) In the introduction, it may be necessary to present the different genes that have been used before for Symbiodinium, and specify their use. Some genes seem to be useful for phylogenetic reconstruction (28S and 23S as mentioned in the manuscript) and others for just a finer genotyping and biogeography (ITS2?). Clarification can be useful in this paper for non-specialists. It would be also useful if authors could explain a bit why 28S is a benchmark gene (resolution or closest match to the biology/biogeography/life style of each clade?)

5) After line 46, authors could say few words about the different characteristics of the nine clades. Do they have specific biogeography? does host spectrum vary in different clades? specialists/generalists? free-living/in hospite? Physiological features?
It will help the non-specialists of coral/Symbiodinium by providing a broad picture on the biology of Symbiodinium.

6) Authors also need to present briefly our current understanding on the divergence pattern and evolution of Symbiodinium. If any, what are the main hypotheses on the Symbiodinium's evolution? Some dates would be informative and insightful. According to Pochon et al 2005, Symbiodinium started to diversify in the Eocene, ca. 50 Ma. So, diversification rate seem to be rather high?

7) In "Material and Methods" section, line 71, it would be useful to give the length of each fragment in brackets.

8) In the "Results" section, line 157, authors found that the concatenation of the six genes provides the same topology as the 28S, but with stronger supports at all nodes. This is an important result and maybe the tree deserves to be included in the main text and not in supplementary. This multi-gene tree is now the updated reference phylogeny of Symbiodinium, which will be widely used. I recommend to include the figure in the main text.

9) line 183-184, this sentence needs to be further explained. This results would mean that irrespective to the organelle, a given clade exhibits the same evolutionary rate compared to the other clades? Is this linked to the sentence line 189-190? Clade A has a slow rate in all genes? a slower rate in mitochondria compared to nuclear and plastidial compartment? Sorry, my personal understanding is not very clear although these analyses are very interesting and are important findings of the paper. Especially, because authors specifically wrote a paragraph on these results.

10) In the "Discussion" section, if possible, I would suggest authors to hypothesize a bit about the sequential evolution in terms of life style/host/biogeography of each clade (after sentence line 223-224). What would explain the slow rate of clade A or high rate of I? their host spectrum/biogeography? They only discuss about the clade D and I but not the other clades.

11) line 228 "architecture" instead of structure?

12) Maybe, it would be interesting to discuss the paper of Roy-Smith and Keeling 2012 (J. Eukaryot. Microbiol., 59(2).) saying "...........for the few lineages in which they have been explored, including land plants and green algae, the mitochondrial DNA mutation rate is nearly always estimated to be lower than or equal to that of the plastid DNA. Here, we show that in protists from three distinct lineages with secondary, red algal-derived plastids, the opposite is true: their mitochondrial genomes are evolving 5–30 times faster than their plastid genomes, even when the plastid is nonphotosynthetic".
So, the study of Pochon et al on the red-plastid-bearing Symbiodinium contrasts with the results of Roy-Smith and Keeling (only on 3 species), and has the same pattern as land plants, green algae etc.. The last sentence of the work of Roy-Smith and Keeling is "It will be interesting to see whether the relative rates of mtDNA vs. ptDNA from other lineages with secondary red plastids show similar trends to those described here".

Would it be possible to explain the slower evolutionary rate of mitochondrial genes by the symbiotic lifestyle of Symbiodinium? Energetic demand of Symbiodinium would be less critical as it can be supplemented by the host? so the mutation pressure on the mitochondrial genome is lower? maybe it is too speculative but it may deserve some hypothetical explanations as it opens new perspectives and paves the way for more analyses. Authors may also read the work of Michael Lynch (e.g. 2006, Science).

13) In the Discussion, it would be also worthwhile to mention the paper of Shaked and de Vargas 2006 (MEPS), showing that the 28S rRNA of Symbiodinium evolves around 6 times faster than other dinoflagellates (including symbiotic dinoflagellates in plankton), except for clade A. It means that in general Symbiodinium clades evolve relatively faster (except A).( and then Pochon et al demonstrate that within Symbiodinium, clades have different evolutionary rates).

14) Sentence 270-271: you could also cite this paper (Decelle 2013 Commun Integr Biol. 1;6(4):e24560. doi: 10.4161/cib.24560) www.landesbioscience.com/journals/cib/article/24560/
Evolutionary rates between Symbiodinium and pelagic symbionts are compared and discussed. Some working hypotheses try to explain the link between evolutionary rates and the life cycle/ importance of the free-living vs symbiotic stage of microalgal symbionts. In plankton, microalgal symbiont seem to evolve slower than Symbiodinium because their free-living stage would be more prevalent than the one of Symbiodinium. In other words, planktonic symbionts experience more free-living settings compared to Symbiodinium that mainly deals with symbiotic settings in hosts. It could be also true for the clades of Symbiodinium?

15) Last minor suggestion, Figure 2 is beautiful and well presented. However, it would be easier for the reader to see the support values (especially the highest) bigger in size. Alternatively, black dots can represent nodes with >95% BP and BiPP of 1.0, so readers could rapidly see the supported clades in the different single-gene phylogenies.

·

Basic reporting

1. The submission should disclose the financing which made their work possible.
2. Please pay attention to the "Basic Manuscript Organization", the title and any legends should be on the same page as the figure or table.

Experimental design

The submission should clearly define the research question, which must be relevant and meaningful, in the introduction.

Validity of the findings

The conclusions should be appropriately stated, should be connected to the original question investigated.

Additional comments

The manuscript “Multi-gene analysis of the symbiotic and free-living dinoflagellate genus Symbiodinium” present a multi-gene analysis of Symbiodinium. Individual and concatenated phylogenies of six functional genes from three cellular compartments were conducted for nine clades of Symbiodinium. And the evolutionary rates were analyzed and compared among the three cellular compartments (nucleus, mitochondria and chloroplast), as well as all known clades and sub-clades. This paper is of great interest to read from its results of the evolutionary rate analysis to the assession of the adaptation of this enigmatic dinoflagellate. I have a few comments/revisions which should be addressed and are listed below. My recommendation is to accept this article after the revisions.
1. Multi-gene analysis supports nr 28S as a benchmark lineage marker, none were evolutionarily identical. However, the genes from three cellular compartments could be useful in genome-wide evolutionary analysis, because of their distinct evolutionary constraints and potential. So, the authors should clearly define the research questions and the strength of the multi-gene analysis.
2. Why do you choose nuclear genes elf2 as one of the target for the evolutionary analysis of the symbiotic and free-living dinoflagellate genus Symbiodinium ?
3. The notes should be added to Figure 2, including the means of a, b, and c, and the numbers on each clusters.

Reviewer 3 ·

Basic reporting

The manuscript adheres to the format requirement of the journal. The only shortcoming in this category is the lack of a specific taxonomic/phylogenetic question to be addressed other than that there is currently a lack of a multi-gene phylogeny for Symbiodinium.

Perhaps determining basal lineages of Symbiodinium can be the question to be addressed in the paper?

Experimental design

Given that increasing use of ITS in Symbiodinium molecular taxonomy, it is a major shortcoming that this gene was not included in the analysis.

While the title promises inclusion of free-living Symbiodinium, there were no explicitly indicated free-living strains in the paper. Furthermore, it is unclear if the strains used in this study are or will be made publicly available upon publication of this paper.

Validity of the findings

A consensus of all the trees is the basal position of clades A, E, D, and G, which is the most interesting result in the paper to me. Perhaps this can be the key question to be addressed in the paper.

Additional comments

The paper can be substantially improved by developing a specific question and using the phylogenetic analysis to address it. ITS should be included or reason should be given why it should/can not be included. Title should be revised unless true free-living strains are identified explicitly among the strains used.

---

## Round 0.2 · accepted · Accept

· Academic Editor

Accept

I am happy to accept your ms for publication in its present form.

Reviewer 1 ·

Basic reporting

I read carefully the revised version of the manuscript. As authors modified the manuscript according to the comments and suggestions of the reviewers (including myself), I accept the manuscript.

Experimental design

No comments

Validity of the findings

No comments